# Fast and Robust Object Tracking Using Tracking Failure Detection in Kernelized Correlation Filter

**Jungsup Shin, Heegwang Kim**  **, Dohun Kim and Joonki Paik** *

Graduate School of Advanced Imaging Science, Multimedia and Film Chung-Ang University, Seoul 06974, Korea; neverbeforepossible@gmail.com (J.S.); heegwang27@gmail.com (H.K.); kdh6126@gmail.com (D.K.)
* Correspondence: paikj@cau.ac.kr

**Abstract:** Object tracking has long been an active research topic in image processing and computer vision fields with various application areas. For practical applications, the object tracking technique should be not only accurate but also fast in a real-time streaming condition. Recently, deep feature-based trackers have been proposed to achieve a higher accuracy, but those are not suitable for real-time tracking because of an extremely slow processing speed. The slow speed is a major factor to degrade tracking accuracy under a real-time streaming condition since the processing delay forces skipping frames. To increase the tracking accuracy with preserving the processing speed, this paper presents an improved kernelized correlation filter (KCF)-based tracking method that integrates three functional modules: (i) tracking failure detection, (ii) re-tracking using multiple search windows, and (iii) motion vector analysis to decide a preferred search window. Under a real-time streaming condition, the proposed method yields better results than the original KCF in the sense of tracking accuracy, and when a target has a very large movement, the proposed method outperforms a deep learning-based tracker, such as multi-domain convolutional neural network (MDNet).

**Keywords:** visual tracking; correlation filter; real-time tracking; multi-domain convolutional neural network (MDNet)

## 1. Introduction

Visual object tracking is one of the most active research topics in the computer vision field because of its wide application areas including: human–computer interaction, autonomous vehicle vision, visual surveillance, and robot vision, etc. In recent years, researchers have paid significant attention to correlation filter (CF)-based tracking algorithms [1–3]. The major advantage of the CF-based tracker is twofold: (i) computational efficiency by replacing the spatial correlation estimation process with element-by-element multiplication in the Fourier transform domain and (ii) being an online learning process that updates the position of the target object every frame.

Despite a lot of studies in the visual tracking community, there are still several challenges such as appearance distortion, light change, rapid motion, motion blur, background clutters, deformation, occlusion, and rotation. Since the trackers trace the target based on the appearance and position of the target in the previous frames, tolerance against appearance and position of the target is a major factor which determines the accuracy of a tracker. Such challenging scenes result in significant mismatch between appearance of the real target and input object, which consequently yields the tracking failure. In this context, deep feature-based trackers have recently attracted increasing attention. Many convolutional neural network (CNN)-based trackers have been proposed to achieve higher accuracy than traditional CF-based trackers [4,5]. The CNN-based trackers outperform CF-based trackers because deep features provide higher tolerance against challenging scenes.

In spite of many advantages from the deep features, CNN-based trackers need to be trained using a very large dataset. Moreover, an excessive computational load is another disadvantage of the CNN-based trackers, which can process less than five frames per second using a personal computer with a single GPU. When real-time tracking is required, computational efficiency becomes the most important issue. Recently, a high performance camera with a frame rate up to 100 FPS or higher has been used for various applications, such as smart phones, action cams, and drones. In a video captured by such high-FPS cameras, appearance variation over frames are reduced whereas the high processing speed becomes more important. Hamed Kiani Galoogahi et al. presented that CF based trackers outperform deep feature based trackers not only in speed but also in accuracy under high FPS circumstances [6]. Furthermore, the use of one or more GPUs for a CNN-based tracker is another burden of a real-time, economic realization of practical video object tracking.

In this paper, we propose a novel kernelized correlation filter (KCF)-based tracking method that improves tracking accuracy under challenging circumstances for high frame rates and real-time tracking. The proposed tracking algorithm processes each frame using three steps: (i) detection of tracking failure by analyzing the peak and average of neighboring correlation values, (ii) re-tracking the target using a tracking failure handing algorithm when the tracking failure is detected, and (iii) calculating a motion vector of the target for selecting preferred search window when tracking failure is detected.

## 2. Related Work

Various tracking approaches have been proposed to solve challenging problems, such as background clutters, partial occlusion, shape deformation, temporary disappearance, illumination changes, etc. [7–9]. Those trackers can be categorized based on several criteria such as on- or off-line training, handcrafted feature- or deep feature-based, and generative or discriminative model-based, etc. In online training, trackers are being trained with data acquired in the tracking process, while off-line trackers need a dataset for a priori training. To represent a target object, handcrafted feature-based trackers utilize features that are manually defined by researchers, whereas deep feature-based trackers utilize features extracted from a deep neural networks (DNNs). Discriminative model-based trackers distinguish target objects from background in an appropriate way, whereas generative model-based trackers search the best-matching window to trace the target. In this section, we summarize state-of-the-art tracking methods by classifying them into two categories: (i) correlation filter-based and (ii) deep learning-based methods.

Correlation filter (CF) is one of the most popular approaches to visual tracking because of its robustness and computational efficiency. Basic mechanism of CF is to generate a response map which has large values in the region-of-interest (ROI), but small values in other regions. In this context, tracking is carried out by estimating position with the highest response value as the target location. To the best of the authors' knowledge, average synthetic exact filter (ASEF) proposed by Bolme et al. [10] and unconstrained minimum average correlation energy (UMACE) proposed by Mahalanobis et al. [11] were the first correlation filter-based approaches to visual tracking. However, their training requirements are not suitable for online tracking.

Bolme et al. [12] proposed an improved version of ASEF, called minimum output sum of square error (MOSSE)-based tracker. The major contribution of the MOSSE-based tracker is the on-line, adaptive training for appearance changes of the target object. After introduction of the MOSSE filter, CF-based tracking methods drew great attention in the visual tracking community and many modified versions of the CF-based tracker have been proposed in the literature. Henriques et al. proposed circulant structured with kernels (CSK) [1] method that exploits the correlation filter in kernel space, and kernelized correlation filter (KCF) [2] extends the kernel trick. In the KCF mechanism, the tracker generates a response map by multiplying the filter kernel and cropped search window at every frame in the Fourier domain, and then estimates the target position by searching the highet value in the inverse Fourier transformed response map. As the KCF achieved outstanding performance,

researchers tried to improve KCF based tracking method, resulting in many derivational methods. Danelljan et al. [13] tried to handle scale variation by training an adaptive multi-scale correlation filter using HOG feature, and Liu et al. [14] tried to solve the partial occlusion problem using multiple correlation filters. Qi et al. [5] proposed hedged deep tracker (HDT), in which several weak trackers are combined to realize a strong tracker. In other words, target positions are estimated from weak trackers, and a final decision is made by the hedge algorithm that combines weak trackers. Ma et al. [15] proposed long-term correlation tracker (LCT), which is robust against target translation and can perform re-detection if tracking failure occurs. Bertinetto et al. [16] presented sum of template and pixel-wise learners (STAPLE) method using HOG and color histograms for target representation. More recently, Yijin et al. [17] proposed Parallel Correlation Filters, Zhaohui et al. [18] presented a tracker using correlation filter fused with color histogram and Hao et al. [19] proposed a correlation filter-based tracker using enhanced features and adaptive Kalman filter.

Meanwhile, deep learning-based tracking methods are actively proposed in the literature. Nam et al. [20] presented multi-domain network (MDNet) that consists of shared layers and a domain-specific fully connected layer. Target representation is carried out in shared layers, while binary classification is performed in the domain-specific fully connected layer. MDNet has shown outstanding performance in accuracy in a benchmark performed by Galoogahi et al. [6]. Fan et al. [21] proposed structure aware network (SANet) as an extended version of MDNet. SANet utilizes deep features from CNN and RNN based on the particle filter. Recently, Siamese network-based trackers have been proposed. In the Siamese network, similarity is calculated by matching the target template and candidate samples. Held et al. [22] proposed generic object tracking using regression network (GOTURN) based on the Siamese network. More recently, Shuo et al. [23] proposed lightweight Siamese network based tracker which can perform which can be updated during tracking process, and Lijun et al. [24] presented a tracking method based on multi-feature fusion Siamese network with Kalman filter.

In addition, a number of different approaches to visual tracking have been proposed, and those can be evaluated by different standards. Generally, the major advantage of CF-based tracking methods is the high processing speed even without using GPU, whereas deep learning-based trackers achieved higher accuracy at the cost of expensive hardware and high computational load. Therefore, it is still worth improving the CF-based tracking methods since low-cost, real-time application is still ubiquitous in visual surveillance applications and numbers of image processing solutions [25–28]. In the following section, we discuss improvement of KCF-based tracking method by integrating techniques which can handle tracking challenges.

## 3. The Proposed Method

The proposed method consists of three parts; (i) tracking failure detection algorithm, (ii) re-tracking algorithm, and (iii) motion vector analysis algorithm, based on KCF method as shown in Figure 1. Tracking failure detection and motion vector analysis algorithms are performed in every frame, and re-tracking algorithm is performed only when the tracking failure is detected. In this section, we discuss about proposed method in detail.

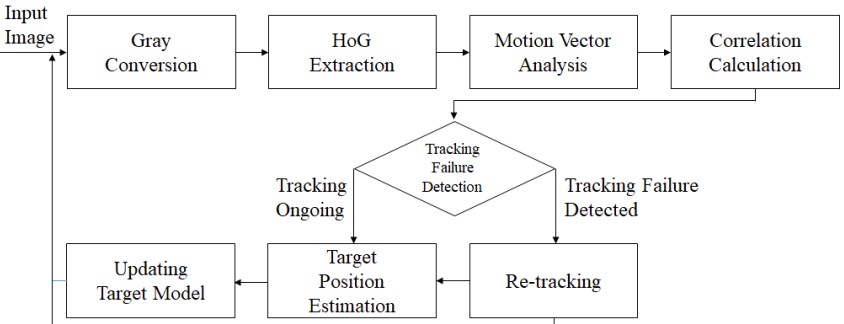

**Figure 1.** Block diagram of the proposed method.

### 3.1. Tracking Failure Prediction by Analyzing Correlation Values

KCF-based trackers calculate correlation values between target and reference using cyclic shift within the pre-specified search window. A correlation value from each cyclic shift corresponds to a cell in the response map. Position of each cell in the response map means the vertical and horizontal shifts. Trackers estimate position of the target by finding the position of a cell with the highest correlation value in the response map and determining the corresponding coordinate in the search window.

Most trackers can successfully track an object if sequences does not include challenging scenes, where there exists a single peak value. As shown in Figure 2a, trackers trace the coordinates of the peak every frame under an ideal tracking condition.

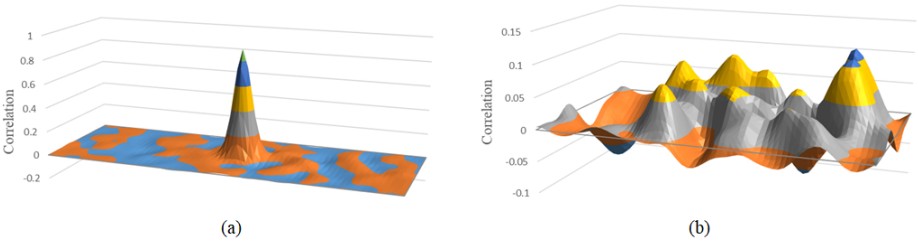

**Figure 2.** (**a**) Common form of correlation distribution from successful tracking sequences. (**b**) An example of correlation distribution from tracking sequences with challenging scenes.

On the other hand, under a challenging circumstance, the peak value relatively decreases and its neighboring values increase as shown in Figure 2b. If a scene contains, for example, a severely blurred target, there is not a single distinct peak correlation value. Since a peak value does not occur at the position of real target in challenging sequences, tracking failure can be predicted by analyzing peak value and neighbor values of the peak.

Let $c_{t(i,j)}$, for $i = 0, 1, 2...n-1$, $j = 0, 1, 2...m-1$, be the correlation value of the response map of size $m \times n$ in $t$th frame. Then, average correlation value of $5 \times 5$ neighboring region around $(p, q)$ is expressed as

$$N_{t(p,q)} = \frac{1}{24}\left(\left(\sum_{i=p-2}^{p+2}\sum_{j=q-2}^{q+2} c_{t(i,j)}\right) - c_{t(p,q)}\right) \tag{1}$$

Since the peak coordinate is $(\bar{p}, \bar{q})$ in the response map, the peak and average correlation values can be expressed as $c_{t(\bar{p},\bar{q})}$ and $N_{t(\bar{p},\bar{q})}$, respectively.

In Equation (1), average value of $c_{t(\bar{p},\bar{q})}$ and $N_{t(\bar{p},\bar{q})}$ over recent $\sigma$ frames can be derived as Equations (2) and (3).

$$P_{\text{aver}} = \frac{1}{\sigma}\sum_{k=t-\sigma+1}^{t} c_{k(\bar{p},\bar{q})} \tag{2}$$

$$N_{\text{aver}} = \frac{1}{\sigma} \sum_{k=t-\sigma+1}^{t} N_{k(\bar{p},\bar{q})} \tag{3}$$

Tracking failure can be predicted by analyzing $P_{\text{aver}}$ and $N_{\text{aver}}$ over frames. In Figure 3, Peak and Neighbor respectively denote $c_{t(\bar{p},\bar{q})}$ and $N_{\text{aver}}$. Figure 3a shows $c_{t(\bar{p},\bar{q})}$ and $N_{\text{aver}}$ over frames for successful tracking cases, where $c_{t(\bar{p},\bar{q})}$ is relatively high and sufficiently distinct from $N_{\text{aver}}$. On the other hand, Figure 3b shows the failure cases, where the $c_{t(\bar{p},\bar{q})}$ significantly dropped, and the difference from $N_{\text{aver}}$ becomes indistinct with some oscillations because of the severe blur. Furthermore, if the peak occurs at a wrong position, the tracking process totally fails.

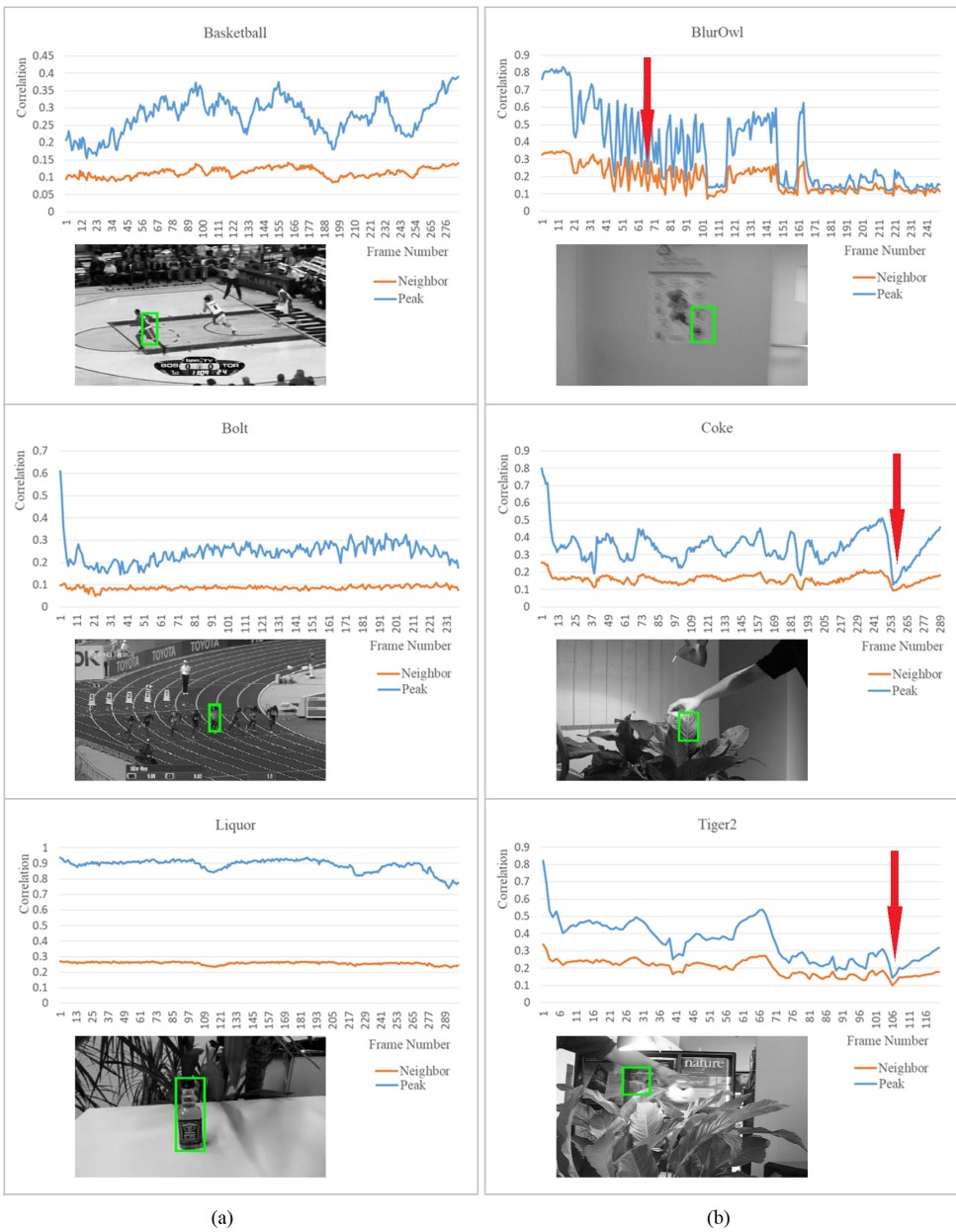

**Figure 3.** Comparison of peak values and average values of neighbor area around the peak. (**a**) Correlation developments with successfully tracked targets. (**b**) Correlation developments in sequences including tracking failures.

To determine when tracking failure occurs, we use a conditional expression such as $\frac{c_{t(\bar{p},\bar{q})}}{P_{\text{aver}}}$ and $\frac{c_{t(\bar{p},\bar{q})}}{N_{\text{aver}}}$, then a conditional expression to decide tracking failure can be defined as

$$\left( \frac{c_{t(\bar{p},\bar{q})}}{P_{\text{aver}}} < T_1 \right) \text{ or } \left( \frac{c_{t(\bar{p},\bar{q})}}{N_{\text{aver}}} < T_2 \right) \tag{4}$$

where $T_1$ and $T_2$ denote thresholds. In (4), we determine the optimal $T_1$ as

$$T_1 = \alpha + \beta \cdot \sigma \tag{5}$$

where, $\sigma$ represents the standard deviation of the peak, and $\alpha$ and $\beta$ be a pair of appropriately chosen constants to increase the accuracy of tracking failure detection. The accuracy versus the $(\alpha, \beta)$ plane is shown in Figure 4.

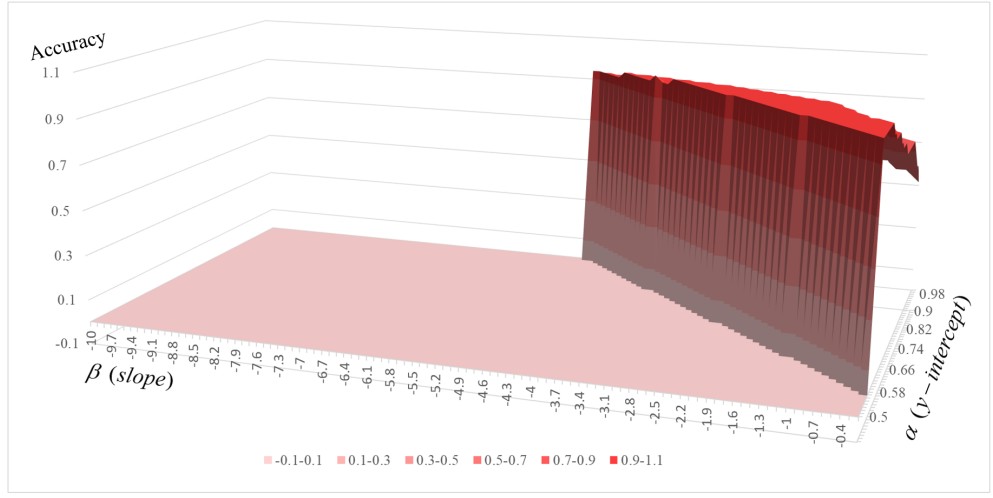

**Figure 4.** Accuracy distribution for distinguishing successfully tracked frames and failed frames according to $\alpha$ and $\beta$.

To derive optimal $\alpha$ and $\beta$, we provided a video sequence containing a frame of tracking failure. We labeled the frame of tracking failure as 'failure' and the others as 'success'. The accuracy of $T_1$ was investigated at every frame for a specific pair of $(\alpha, \beta)$. As shown in Figure 4, accuracy of $(\alpha, \beta)$ is set to zero if the pair cannot detect tracking failure. After experiments using Visual Tracker Benchmark dataset [29], we found that the optimal $\alpha$ is 0.85 and $\beta$ is $-2.4$. Though there are slight differences among video sequences, $(0.85 - 2.4 \cdot \sigma)$ values are around 0.6 and $\frac{c_{t(\bar{p},\bar{q})}}{N_{\text{aver}}}$ values at tracking failure points are around 0.6 as well. Therefore, we performed experiments by setting $T_1$ and $T_2$ as 0.6.

### 3.2. Re-Tracking a Disappeared Target Using Multiple Search Windows

Although it is technically impossible to keep tracking if a target completely disappears in a scene, we assume that the target appears again in the neighborhood of where it disappeared. To re-start tracking a disappeared target, multiple search windows are deployed around the position where the target disappeared. During the re-tracking process, we do not update target appearance. For example, tracking failure due to occlusion is predicted as shown in Figure 5a, then multiple search windows will be deployed as shown in Figure 5b. Response map and peak value is generated from each search window, and the tracker starts tracking again if the target is captured in any of those search windows. We reduced computational loads by giving priority to these search windows, details are explained in the next section. Desired results could be derived by re-tracking the target from a peak point at which condition $\frac{c_{t(\bar{p},\bar{q})}}{P_{\text{aver}}} > 0.6$ is met in any of multiple search windows.

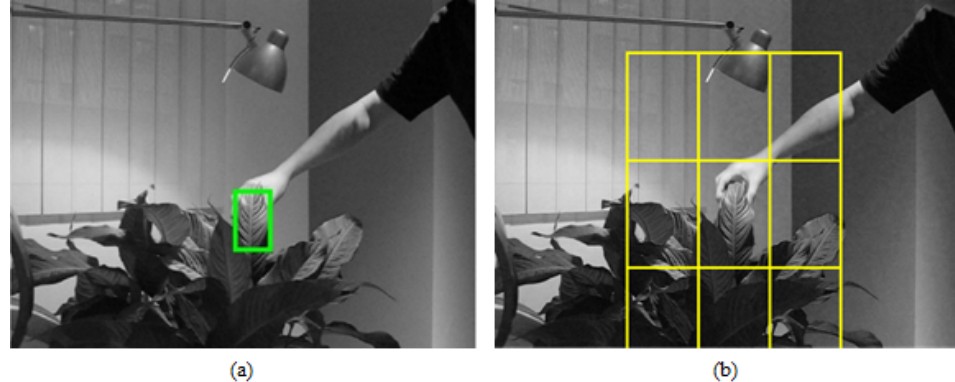

**Figure 5.** (**a**) Tracking failure detected as the target disappeared. (**b**) Multiple search windows are deployed in tracking failure handling process.

### 3.3. Prioritization of Search Windows Based on Motion Vectors

In many practical cases such as watching pedestrians or cars with CCTV, targets show linear movements for a number of frames. It means that movements of targets in the next frame can be predicted by analyzing recent movements of the targets. For this, we analyze movements of the targets by averaging motion vectors of the targets from recent 3 frames as expressed in Equation (6)

$$\frac{1}{3} \sum_{i=n-3}^{n-1} (vx_i, \ vy_i) \tag{6}$$

$vx_i$ and $vy_i$ denote horizontal coordinates changes and vertical coordinates changes of the target in *i*th frame and *n* denotes current frame number.

We separate values of motion vectors being analyzed during tracking into 9 regions as shown in Figure 6a. If absolute values of x and y of a motion vector are smaller than 2, it means there are no significant movements so that the motion vector belongs to region number 5. In other cases, each motion vector belongs to each region based on its direction. And each region of motion vector is associated with each search window as shown in Figure 6b. As tracking failure is detected, response map is generated in the search window which is associated with region of current motion vector of the target first, computational load can be reduced by skipping other search windows if the target is re-captured in the preferred search window.

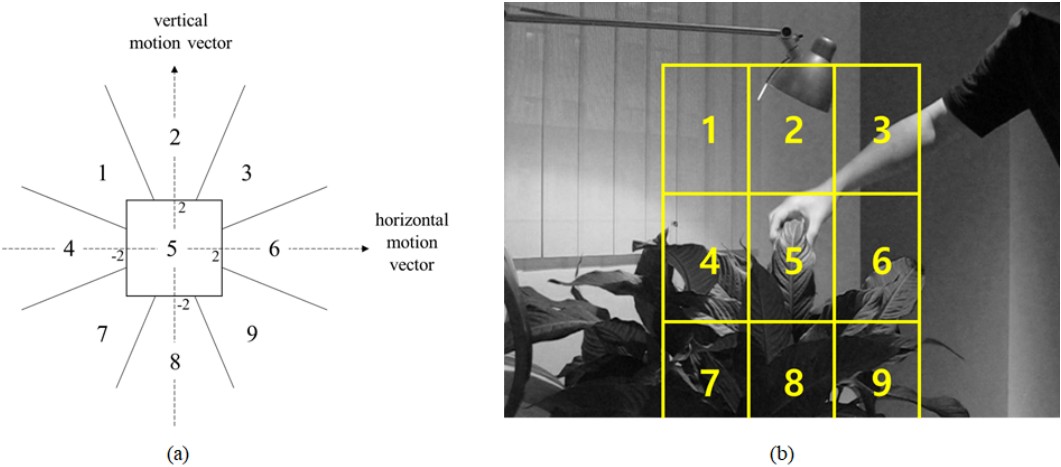

**Figure 6.** (**a**) 9 regions of motion vectors. (**b**) Each search window associated with each region of motion vector.

## 4. Experimental Results

To establish streaming environment, images files in Visual Tracker Benchmark dataset [29] were converted into movie clips in MPEG-4 format. Each image has time stamp interval of 0.03 s so that the converted movie clips are played at 33.3 frame per second as shown in Figure 7.

We acquired additional test videos of routine street scenes using a CCTV over the street, called CAU_Ped1, CAU_Ped2, CAU_Ped3 and CAU_Ped4. Those street scenes are suitable to evaluate the proposed method because some targets are completely occluded by other pedestrians, vehicles and trees very often.

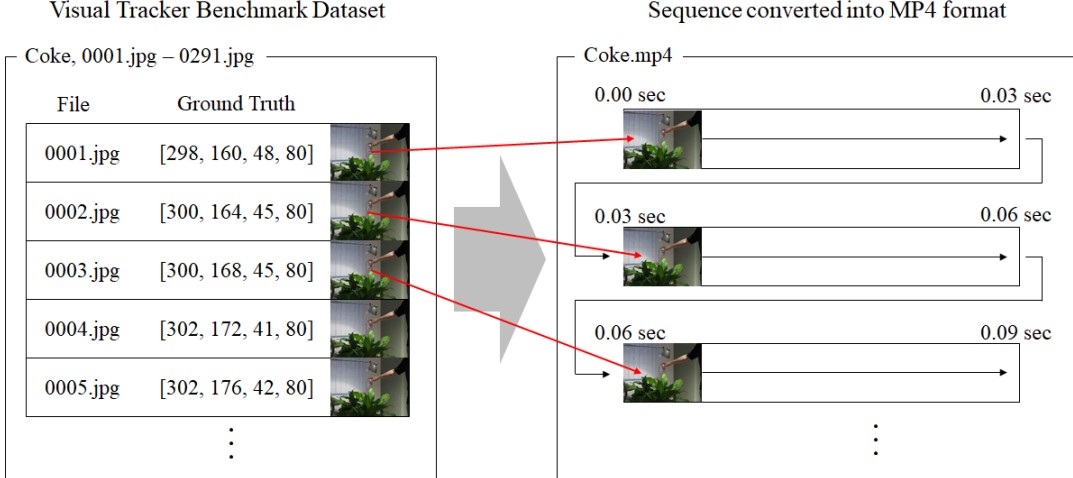

**Figure 7.** Dataset conversion into movie clip of mp4 format.

In image file-based datasets, ground truth is provided for each image as shown in Figure 7. Each value of ground truth denotes *x*-coordinate, *y*-coordinate, width and height of the target. On the other hand, ground truth is provided for a certain period in the streaming condition as shown in Figure 8. Spatial location of the target estimated by the tracker is saved with elapsed time, and evaluated by being compared to the time-based ground truth. In this work, we used a PC equipped with an Intel i7 8700K CPU and GTX 1080ti.

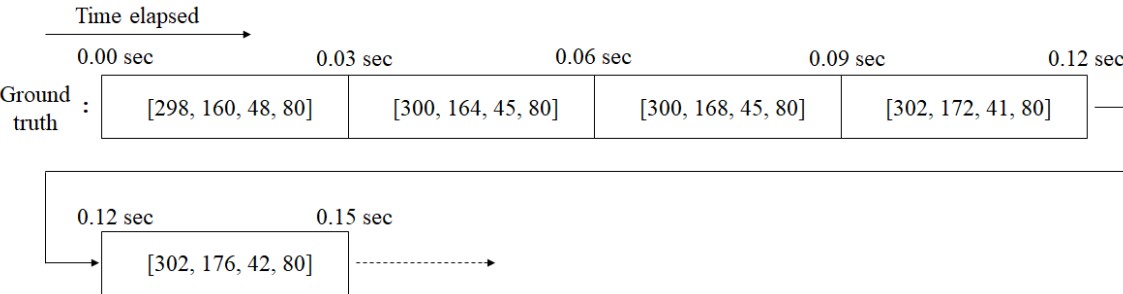

**Figure 8.** Time based ground truth.

### 4.1. Tracking Performance of Deep Feature Based Tracker and the Proposed Method Based Tracker

Galoogahi et al. evaluated handcrafted feature-based trackers and deep feature-based trackers using their own datasets [6]. In that evaluation, MDNet-based tracker provided the highest accuracy at the cost of a slow processing speed, which is a common problem of various deep feature-based trackers. Under the streaming condition, the processing speed of a tracker is directly related to the tracking accuracy. In other word, a slow tracker loses a number of frames for each tracking period, which results in accuracy degradation.

In evaluating the MDNet-based tracker, tracking accuracy was measured under three conditions including (i) local dataset using a GPU, (ii) streaming sequences using a GPU, and (iii) streaming sequences using only CPU. Intersection over Unions (IoUs) are collected from each frame and average of them was used as the final accuracy for each dataset. As shown in Figure 9, the accuracy of tracking was significantly decreased using only CPU under the streaming condition, which demonstrated that the speed of tracker is an important factor to determine the accuracy in the streaming condition.

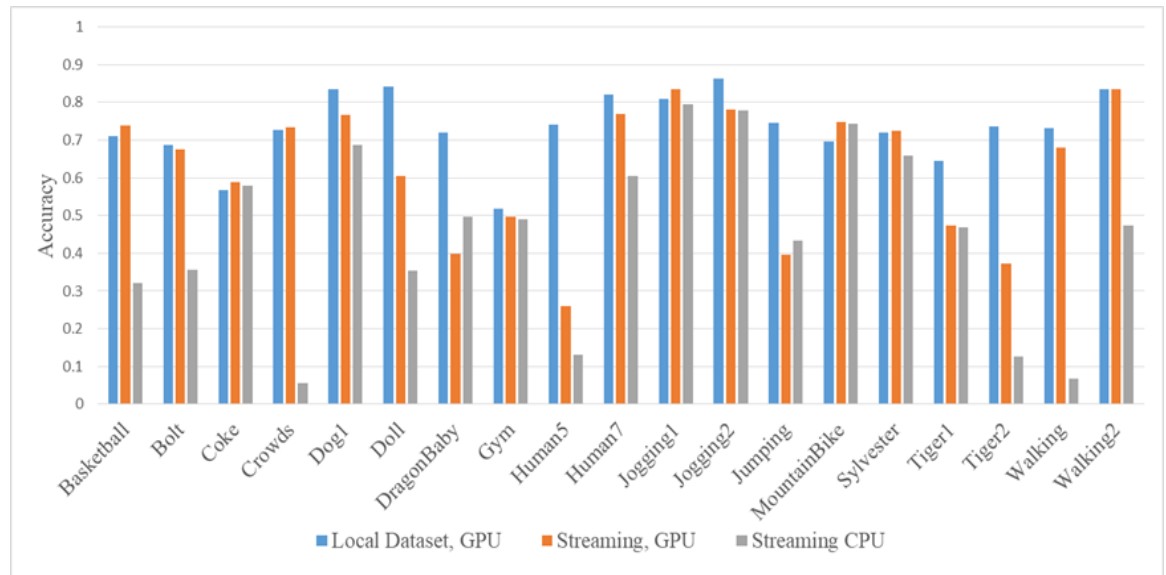

**Figure 9.** Accuracy evaluation of MDNet based tracker under 3 conditions: local dataset with GPU processing, streaming sequences with GPU processing, and streaming sequences with CPU processing.

We compared the accuracy of the proposed method, MDnet-based tracker and GOTURN-based tracker under a streaming condition as shown in Figure 10. More specifically, Figure 10a shows that the proposed method yielded an acceptable accuracy, where the position of targets drastically change and the reduced frame rate degrades the tracking accuracy. Some sequences include challenging scenes such as occlusions or blurring, which requires tracking failure detection and restarting the tracking process after detecting the failure. In CAU_Ped1, CAU_Ped2, CAU_Ped3 and CAU_Ped4, videos recorded by CCTV watching pedestrians on the street, occlusions frequently occur. For that case, the proposed tracker has been proved suitable for those applications.

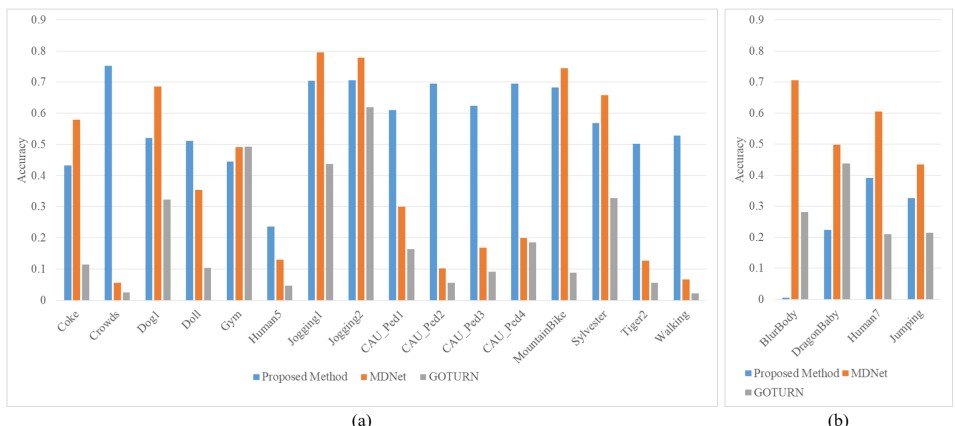

(a)                                      (b)

**Figure 10.** Accuracy evaluation of the proposed method, MDNet-based tracker and GOTURN-based tracker under a streaming condition. (**a**) Datasets in which significant target movements or challenging scenes exist. (**b**) Datasets in which no significant target movements and challenging scenes exist.

On the other hand, deep feature-based trackers can successfully track a continuously blurred and slightly moving target which is still close to the disappearing location as shown in Figure 10b. In that case, the proposed method keeps spending time for the re-tracking process because tracking failures are detected with a degraded accuracy.

In real applications, a low processing speed would directly degrade tracking accuracy because a camera cannot follow the target by itself.

### 4.2. Tracking Performance of the Proposed Method Based Tracker and Original KCF Based Tracker

We found meaningful improvements in the sense of tracking accuracy from evaluations performed using the proposed tracker and the original KCF tracker. As shown in Figure 11a, in evaluation using various datasets, the original KCF tracker failed in tracking while the proposed tracker performed successful tracking and yielded improved tracking accuracy. At least one challenging scene is included in these sequences, the proposed tracker outperformed original KCF tracker because of the re-tracking process. The proposed tracker and the original KCF tracker yielded comparable results in some other datasets in which both trackers have successfully tracked the targets as shown in Figure 11b. Lastly, both of trackers have failed in tracking in datasets named DragonBaby and Jumping where the targets are moving rapidly as shown in Figure 11c.

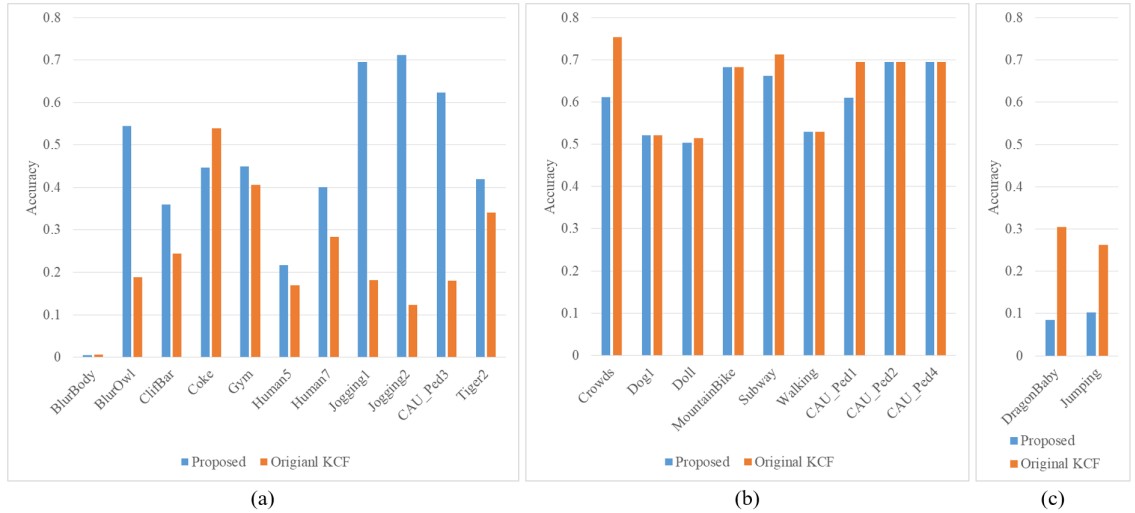

**Figure 11.** Accuracy comparison between the proposed method and original KCF method. (**a**) The proposed method yielded successful tracking while the original KCF based tracker failed. (**b**) Both of the trackers yielded successful tracking. (**c**) Both of the trackers failed.

### 4.3. Re-Tracking Steps on Tracking Failures

Figure 12 shows how the proposed method re-capture the target when tracking failure occurred. Figure 12a shows tracking failures in different sequences. The tracking failure is detected and multiple search windows are deployed as shown in Figure 12b. A red search window out of nine neighboring search windows is shown in Figure 12b, which represents the preferred search window. Tracking process in other search windows can be skipped if the target is captured in the preferred search window. In datasets BlurOwl and Coke, there is no advantage of preferred search windows because of irregular movements of the targets. On the other hand in Jogging1, CAU_Ped1, CAU_Ped3 and Tiger2, there are linear movements of the targets so that computational loads could be reduced as the targets were captured in the preferred search windows. As the targets are captured in the preferred search windows or other search windows, tracking is re-started from the corresponding position as shown in Figure 12c.

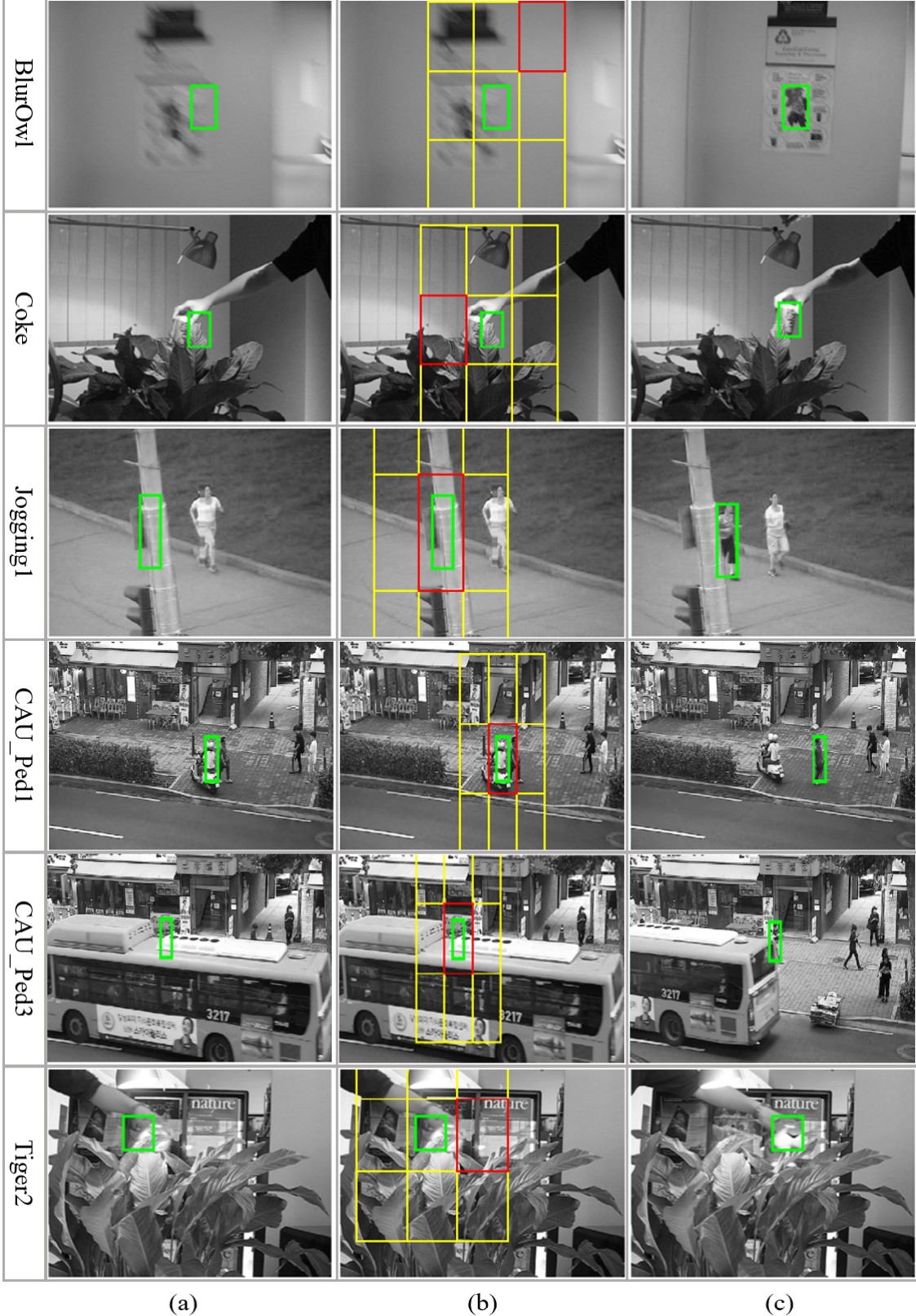

**Figure 12.** Target re-capturing process. (**a**) Tracking failure detected. (**b**) Tracking failure handling. (**c**) Target re-captured.

## 5. Conclusions

In this paper, we presented tracking failure detection and re-tracking algorithms to improve the KCF-based tracker. The goal of this work was to develop a real-time tracking algorithm with improving the tracking accuracy. Performance of the tracking failure detection algorithm has a decisive effects on total performance of the proposed method. More specifically, the proposed method restarts the tracking algorithm when tracking failure is first detected in a simple, effective manner. The proposed algorithm can predict a tracking failure by analyzing the peak and neighboring correlation values with a little extra computational load. Under the tests using the Visual Tracker Benchmark dataset [29] and our own video sets, we precisely found the threshold value for the tracking failure point. Although the re-tracking process requires an additional computational load for multiple search windows, we minimized the load by giving priority to search windows. At this step, appearance of the target can be preserved by skipping update. As a result, the proposed method could compensate for the defect of the original KCF in challenging scenes including occlusions and blurrings. Although many tracking methods exhibit high accuracy under an offline condition, there were few online tracking methods that give a sufficiently high performance. In this context, the proposed method is suitable for a low-cost, real-time tracking application without using a special hardware accelerator such as a GPU. In the future, we will investigate more tracking failure cases, and then develop a corresponding re-tracking algorithm.

**Author Contributions:** Conceptualization, H.K. and J.P.; methodology, software, validation, J.S.; data curation, D.K.; writing—Original draft preparation, Jungsup Shin; writing—Review and editing, H.K.; supervision, project administration, J.P. All authors have read and agreed to the published version of the manuscript.

**Funding:** This work was partly supported by Institute for Information & communications Technology Promotion(IITP) grant funded by the Korea government(MSIT) (2017-0-00250, Intelligent Defense Boundary Surveillance Technology Using Collaborative Reinforced Learning of Embedded Edge Camera and Image Analysis), and by the ICT R&D program of MSIP/IITP [2014-0-00077, development of global multi-target tracking and event prediction techniques based on real-time large-scale video analysis].

**Conflicts of Interest:** The authors declare no conflict of interest.

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
