# Peer review of "Fast and Robust Object Tracking Using Tracking Failure Detection in Kernelized Correlation Filter"

_applsci, doi:10.3390/app10020713_

Round 1

Reviewer 1 Report

This paper shows a good introduction of improved object tracking algorithm. However, I would like to see this paper published after considering some following comments.

The paper obviously needs to be fitted to the article standards excluding i.e. 'journal not specified' note in the footnote of the manuscript.

Some specific comments:

Line 12-13 (abstract): Such a sentence should be avoided if many of your results are opposite

Line 153: Can you provide more details about the experiment?

Figure 7: Please, explain what means ground truth values. Coordinates in the image?

Line 192: The figure lacks embedding in the text

Line 216-218: How can you explain such low accuracy of proposed method for challenging images?

Line 231: I assume that the term 'that' or 'how' is not necessary in this sentence

Author Response

Thank you.

Reviewer 2 Report

This work describes some modifications to the Kernelized Correlation Function (KCF) method of tracking to allow for more robust performance in the light of dropped frames or occlusions. The presented improvements seem to be a fairly incremental advance on the KCF technique, involving essentially a test of the compactness of the correlation peak and then a prior motion vector guided window search for the next content. The proposed algorithm is tested against the state of the art and the results do show improvement over previous methods, even though the originality of the proposed method doesn't seem particularly strong over the existing KCF technique. The proposed method is novel, but it would be nice if the advance on previous method was more than checking for a reduction of correlation peak strength and searching for the target along a direction related to its last known motion. This seems a fairly obvious incremental advance to me. I suggest the authors add some text to clearly describe better the novelty of this work.

Author Response

Thank you.

Reviewer 3 Report

Dear authors,

The paper is of high quality and I recommend its publication after minor revisions:
- Split the first paragraph in Introduction to more, add references to relevant literature,
- Line 30: replace "has" with "have",
- Line 46: I believe that "rates" (plural) and replacing the comma with "and" would be more appropriate here,
- Lines 52-63: add references to relevant literature to support the claims,
- Section 2: split the second paragraph to more,
- Equations 2 and 3: "aver" is not a variable, but only a subscript denoting "average" - it should not be in italics,
- Lines 139-142: the sentence is too long and complicated,
- Figure 3: it would be nice to show the images (objects) that had been tracked,
- Equation 5: I suggest replacing SDoP by a more elegant notation,
- Line 152: alpha and beta are not zero in Figure 4,
- Line 177: ARE no significant movements,
- Line 194: remove redundant "and".

Author Response

Thank you.
